# Impact of fatigue as the primary determinant of functional limitations among patients with post-COVID-19 syndrome: a cross-sectional observational study

Sarah Walker ©,[1] Henry Goodfellow ©,[2] Patra Pookarnjanamorakot,[3] Elizabeth Murray ©,[4] Julia Bindman,[5] Ann Blandford ©,[6] Katherine Bradbury,[7] Belinda Cooper,[5] Fiona L Hamilton,[5] John R Hurst ©,[8] Hannah Hylton,[9] Stuart Linke,[5] Paul Pfeffer,[9] William Ricketts ©,[10] Chris Robson,[11] Fiona A Stevenson,[5] David Sunkersing,[5] Jiunn Wang,[12] Manuel Gomes,[12] William Henley ©,[1] Living With Covid Recovery Collaboration[13]

For numbered affiliations see end of article.

**Correspondence to**
Dr William Henley;
w.e.henley@exeter.ac.uk

## ABSTRACT

**Objectives** To describe self-reported characteristics and symptoms of treatment-seeking patients with post-COVID-19 syndrome (PCS). To assess the impact of symptoms on health-related quality of life (HRQoL) and patients' ability to work and undertake activities of daily living.

**Design** Cross-sectional single-arm service evaluation of real-time user data.

**Setting** 31 post-COVID-19 clinics in the UK.

**Participants** 3754 adults diagnosed with PCS in primary or secondary care deemed suitable for rehabilitation.

**Intervention** Patients using the Living With Covid Recovery digital health intervention registered between 30 November 2020 and 23 March 2022.

**Primary and secondary outcome measures** The primary outcome was the baseline Work and Social Adjustment Scale (WSAS). WSAS measures the functional limitations of the patient; scores of ≥20 indicate moderately severe limitations. Other symptoms explored included fatigue (Functional Assessment of Chronic Illness Therapy–Fatigue), depression (Patient Health Questionnaire–Eight Item Depression Scale), anxiety (Generalised Anxiety Disorder Scale, Seven-Item), breathlessness (Medical Research Council Dyspnoea Scale and Dyspnoea-12), cognitive impairment (Perceived Deficits Questionnaire, Five-Item Version) and HRQoL (EQ-5D). Symptoms and demographic characteristics associated with more severe functional limitations were identified using logistic regression analysis.

**Results** 3541 (94%) patients were of working age (18-65); mean age (SD) 48 (12) years; 1282 (71%) were female and 89% were white. 51% reported losing ≥1 days from work in the previous 4 weeks; 20% reported being unable to work at all. Mean WSAS score at baseline was 21 (SD 10) with 53% scoring ≥20. Factors associated with WSAS scores of ≥20 were high levels of fatigue, depression and cognitive impairment. Fatigue was found to be the main symptom contributing to a high WSAS score.

### STRENGTHS AND LIMITATIONS OF THIS STUDY

⇒ Large cohort of patients (n=3754) with novel disease from 31 specialised post-COVID-19 clinics in England and Wales.

⇒ Patient-reported outcome measures (PROMs) contain eight validated questionnaires including common post-COVID-19 syndrome (PCS) symptoms, quality of life (EQ-5D) and functional status (Work and Social Adjustment Scale), allowing comparison with other health conditions.

⇒ High completion rate of PROMs at baseline (registration) ensures reported data are representative of Living With Covid Recovery digital health intervention (DHI) users.

⇒ As data were collected through a DHI, some clinical data on patients with PCS were not available, such as date of acute COVID-19 infection(s) and vaccination status.

⇒ Regression analysis was used on available data; we acknowledge that missing data may have introduced bias.

**Conclusion** A high proportion of this PCS treatment-seeking population was of working age with over half reporting moderately severe or worse functional limitation. There were substantial impacts on ability to work and activities of daily living in people with PCS. Clinical care and rehabilitation should address the management of fatigue as the dominant symptom explaining variation in functionality.

### INTRODUCTION

Post-COVID-19 syndrome (PCS), or 'long COVID', is defined by National Institute for Health and Care Research and the WHO as the signs and symptoms of the disease that

continue for more than 12 weeks after the initial acute COVID-19 infection.[1] It is causing increasing concern due to the potential number of patients infected and the associated morbidity caused by the symptoms.

As of the second August 2022, there have been over 577 million cases of COVID-19 worldwide.[2] There have been various estimates on the number of patients with acute COVID-19 that go on to develop PCS, ranging from 3.0% to 14.1%[1 3–6] with over 1.4 million people in the UK reporting PCS symptoms as of July 2022.[6] The symptoms of PCS include fatigue, breathlessness, brain fog, anosmia and mental health problems. These symptoms can cause debilitating functional and psychological limitations[3 7] and have been shown to persist for up to 2 years.[1 3 6 8–10] This has led to many people with PCS being unable to work or care for others for a prolonged period.[7] The potential impact of PCS on national health services, economies and population health is attracting international attention as the associated morbidity and economic effects become clearer.[5 11–17]

The UK National Health Service (NHS) has set up post-COVID-19 assessment clinics to provide care for the large number of patients with PCS.[6 18] In the absence of pharmacotherapies shown to be effective for this condition, management of people with PCS has to date focused on self-management education and rehabilitation programmes. These clinics provide specialist rehabilitation from a range of healthcare professionals including respiratory specialist doctors, general practitioners, physiotherapists, occupational therapists and psychologists. Over 30 of these clinics were augmented with a bespoke digital health intervention (DHI), called Living With Covid Recovery (LWCR), to enable remote rehabilitation for patients with PCS during the COVID-19 pandemic. Internationally, despite the growing number of patients with PCS, the strategies to combat PCS are at their early stages with no standard rehabilitation pathway.[11–14] As the pandemic continues, PCS will continue to add significant workload for health services beyond acute COVID-19 care.[19]

This study is the first to present the baseline symptoms and functional impairment from a treatment-seeking PCS population across multiple centres and to estimate the contribution of different patient-reported symptoms to impairment. These data will help clinicians and policy makers plan appropriate services.

## METHODS
### Design and setting
This is a cross-sectional observational study of patients using the LWCR DHI as part of their assessment and treatment in 31 self-selecting specialised post-COVID-19 clinics in England and Wales.

### Intervention
LWCR is a bespoke DHI designed to be part of post-COVID-19 clinics. The LWCR DHI was designed by a multidisciplinary team of clinicians, patient and public involvement (PPI), academics and industry partners.[20] The product was first launched in a clinical setting in August 2020 and since then has been updated eight times. The DHI contains 12 (8 validated) patient-reported outcome measures (PROMs) in the form of validated questionnaires completed by patients as part of their clinical care. In this study, we use 10 of these (8 validated). Six are related to symptoms and one was related to each of patient demographics (unvalidated), functional ability, quality of life and health service use (unvalidated). More details are provided in the Patient-reported outcome measures section and in the study protocol. The Work and Social Adjustment Scale (WSAS) questionnaire was introduced in February 2021 and the demographic questionnaire in April 2021. Development followed the principles of human computer interaction agile development, with updates to the DHI based on feedback from healthcare practitioners and our PPI group. All data collected in the LWCR product were pseudo-anonymised using a unique patient ID number and were stored in Metabase ( www.metabase.com).

### Population
Patients included in this study were those who had registered to use the LWCR DHI as part of the clinical care provided in a PCS NHS community clinic in England and Wales. Patients are referred to these clinics from primary or secondary care after having experienced post-COVID-19 symptoms for 12 weeks or more.

Eligible patients were identified as being suitable for remote rehabilitation service by the clinic if they were aged 18 or over, had access to a smart phone device, were considered likely to benefit from the intervention, were fit for rehabilitation and were able to read English. Patients registered on the LWCR DHI between 30 November 2020 and 23 March 2022.

### Outcomes
#### Primary outcome
The WSAS was the primary outcome measure for this study. WSAS is a validated questionnaire for functional impairment.[21] Scores range between 0 and 40, with scores of 20 or more indicating moderately severe or worse impairment on daily functioning.[21] The WSAS contains five equally weighted component scores (range 0–8) relating to impairments across the following domains:
► Ability to work.
► Home management.
► Social leisure activities.
► Private leisure activities.
► Close relationships.

Additionally, there is a further question to identify those individuals who are either retired or have chosen not to work. There is no defined recall period for the WSAS; therefore, the questionnaire reflects the current situation.

## Secondary outcome

The secondary outcome was the EQ-5D, a standardised measure of health-related quality of life (HRQoL).[22] The EQ-5D-5L descriptive system comprises five dimensions (mobility, self-care, usual activities, pain/discomfort and and anxiety/depression). For each dimension, there are five possible responses (level 1: no problems, level 2: slight problems, level 3: moderate problems, level 4: severe problems and level 5: unable to/extreme problems). The responses are coded to give a five-digit code to describe the respondent's health state (such as 13254). Reference weights from the UK general population are applied to the resulting health states to produce a single summary index score for health status, the EQ-5D-5L index score. This is a measure anchored at 0 (representing 'death') and 1 ('full health'), but it can include negative values to reflect health states judged worse than death. Similar to the WSAS, there is no recall period defined for the EQ-5D; therefore, the PROM would reflect the health status on the day of questionnaire completion.

## Explanatory variables

### Patient demographics

The data collected in the Patient Demographic Questionnaire included patient-reported age, gender, ethnicity, highest level of education and postcode. Patient age and gender were also reported by the clinic when registering the patient to use the DHI. Early versions of the DHI did not include the demographic questionnaire, which became available to all patients in April 2021. Where both clinic and patient-reported data were available, patient-reported age, gender and ethnicity were used, with clinic-reported data used as back-up.

To keep the data pseudo-anonymised, the Index of Multiple Deprivation (IMD) was provided to the study statistician rather than the patient postcode. The English Indices of Deprivation (2019) were used to provide the IMD from the patient's postcode.[23] The IMD decile was not provided for 35 patients who had completed the demographic questionnaire. These were either entered incorrectly or were new, so these were not in the latest update of the IMD registry. Additionally, patient date of birth (as supplied by the clinic) was replaced with year of birth, from which an approximate age could be calculated.

### Patient-reported outcome measures

In this study, six validated questionnaires were used to capture the severity of five of the core symptoms of PCS through PROMs. The PROMs were completed by patients based on their clinical need, as determined by the patients themselves or with their healthcare professional. The first PROM completed by the patient was taken as their baseline measurement. The date and time of completion in relation to when the patient first registered to use the DHI were recorded, along with the outcome scores. PROMs were analysed as continuous variables, unless stated otherwise. Where threshold values for caseness are

available, we present the number of patients within each of these categories to enable comparison between this study and other research.

► Breathlessness.
  Dyspnoea-12 gives an overall score of breathlessness impact, with higher scores corresponding to greater severity[24–26] (recall period not defined, reflects current moment).
  Medical Research Council (MRC) Dyspnoea Scale measures the degree of breathlessness related to activity, with higher scores corresponding to greater severity.[27 28] The scale takes values 1–5 using the following classifications: MRC 1 (mild), MRC 2–3 (moderate) and MRC 4–5 (severe).[29] We analysed this variable as a categorical score (recall period not defined, reflects current moment).

► Fatigue. Functional Assessment of Chronic Illness Therapy–Fatigue (FACIT-F) measures self-reported fatigue and its impact on daily activities and function with lower scores corresponding to greater fatigue. A threshold value of 30 was chosen in line with fatigue reported in a cancer population.[26] Population mean value for FACIT-F in the general population has been reported as 43[25 26 30] (recall period: 7 days).

► Anxiety. The Generalised Anxiety Disorder Scale, Seven-Item (GAD-7) is used as a screening tool and severity measure for anxiety.[31] A cut-off value of 10 or more identifies anxiety. Additionally, threshold values are also considered: no anxiety, 0–4; mild anxiety, 5–9; moderate anxiety, 10–14; and severe anxiety, 15–21 (recall period: 2 weeks).

► Cognition (brain fog). The Perceived Deficits Questionnaire, Five-Item Version (PDQ-5) measures the degree to which individuals perceive themselves as experiencing cognitive difficulties.[32 33] Higher scores indicate more perceived deficits. The following threshold values suggested by Lam[34] are used: minimal, 0–8; moderate, 9–14; and severe 15–20 (recall period: 4 weeks).

► Depression. The Patient Health Questionnaire–Eight Item Depression Scale (PHQ-8) was chosen over the Patient Health Questionnaire–Nine Item Depression Scale (PHQ-9) PROM for this study as it was not always certain that adequate intervention would be available if the question on suicidal thoughts or self-harm was endorsed; therefore, this question was omitted.[35] The same scoring thresholds are used as for PHQ-9, with a score of 10 or more used as a cut-off for a diagnosis of depression[36] (recall period: 2 weeks).

## Statistical analysis

### Primary outcome

Logistic regression was used to identify the PROMs associated with a high WSAS score ($\geq$20) after accounting for the effects of demographic variables. First, we built a model for the demographic factors associated with high WSAS score. Age and gender were included as covariates in all models. Other demographics, including highest level of

**Table 1** Sociodemographic characteristics of the patients in the study

| Patient characteristic n (%), unless stated otherwise | Study population N (%) (N=3754) | WSAS completed n (%) (n=2627) | EQ-5D-5L completed n (%) (n=2643) |
|---|---|---|---|
| Age (years), mean (SD) | 47.7 (12.3) (n=3753) | 47.2 (11.9) | 47.2 (11.9) |
| Age category (years) | | | |
| 18–29 | 349 (9.3) | 236 (9.0) | 237 (9.0) |
| 30–39 | 615 (16.4) | 439 (16.7) | 440 (16.6) |
| 40–49 | 1084 (28.9) | 771 (29.3) | 773 (29.2) |
| 50–59 | 1127 (30.0) | 815 (31.0) | 820 (31.0) |
| 60–69 | 469 (12.5) | 310 (11.8) | 317 (12.0) |
| 70 and over | 109 (2.9) | 56 (2.1) | 56 (2.1) |
| Missing* | 1 | 0 | 0 |
| Gender | | | |
| Female | 2675 (71.3) | 1898 (72.3) | 1909 (72.3) |
| Male | 1060 (28.2) | 719 (27.4) | 724 (27.4) |
| Non-binary | 10 (0.3) | 9 (0.3) | 9 (0.3) |
| Missing* | 9 | 1 | 1 |
| Highest educational level | | | |
| No education | 113 (4.1) | 106 (4.1) | 102 (4.0) |
| School leaver (NVQ 1–2) | 611 (22.1) | 574 (22.5) | 574 (22.6) |
| A-level (NVQ-3) | 574 (20.8) | 532 (20.8) | 533 (21.0) |
| Degree (NVQ-4) | 581 (21.0) | 527 (20.6) | 526 (20.7) |
| Postgraduate degree (NVQ-5) | 885 (32.0) | 817 (32.0) | 808 (31.8) |
| Missing* | 990 | 71 | 100 |
| Ethnicity | | | |
| White | 2414 (87.3) | 2242 (87.7) | 2234 (87.8) |
| Asian or Asian British | 177 (6.4) | 159 (6.2) | 155 (6.1) |
| Black African Caribbean or black British | 55 (2.0) | 48 (1.9) | 47 (1.8) |
| Mixed or multiple ethnicity | 67 (2.4) | 61 (2.4) | 62 (2.4) |
| Other ethnic group | 32 (1.2) | 27 (1.1) | 26 (1.0) |
| Prefer not to say | 19 (0.7) | 19 (0.7) | 19 (0.7) |
| Missing* | 990 | 71 | 100 |
| IMD quintile | | | |
| 1–2 (20% most deprived) | 289 (10.6) | 274 (10.9) | 272 (10.8) |
| 3–4 | 537 (19.7) | 500 (19.8) | 491 (19.6) |
| 5–6 | 657 (24.1) | 610 (24.2) | 606 (24.1) |
| 7–8 | 604 (22.1) | 555 (22.0) | 556 (22.1) |
| 9–10 (20% least deprived) | 642 (23.5) | 585 (23.2) | 586 (23.3) |
| Missing* | 1025 | 103 | 132 |

*Data on patient-reported characteristics are missing for 990 who did not complete the Patient Demographics Questionnaire. In addition, a further 35 are missing IMD as their IMD decile was not available. Percentages do not include those with missing values in the denominator.
IMD, Index of Multiple Deprivation; NVQ, National Vocational Qualification; WSAS, Work and Social Adjustment Scale.

education, ethnicity (as white or non-white) and IMD quintile, were added using a stepwise approach based on the Likelihood Ratio (LR) Test. Any demographic variables with a p value below 0.2 were retained for inclusion in subsequent models. At each stage, the McKelvey and Zavoina's $R^2$ value of the model including the additional

term was calculated as a measure of the proportion of variation in the binary WSAS outcome attributable to the selected factors.[37]

The FACIT-F score was reversed (calculated as 52 minus the reported score) to align the direction of the score with other variables in the analysis. Higher values of the score now represent greater fatigue. We refer to this as FACIT-F (Reversed Scale).

Next, we added each of the PROMs (Dyspnoea-12, MRC-Dyspnoea, FACIT-F (Reversed Scale), GAD-7, PDQ-5 and PHQ-8) in a univariable fashion to the logistic regression model for the demographic factors. Any PROMs with a p value below 0.2 were retained for potential inclusion in subsequent models. A multivariable model including both demographics and PROMs was developed by sequentially adding or removing PROMs according to the LR Test using a p value threshold of 0.05. McKelvey and Zavoina's $R^2$ value was calculated at each stage as a measure of model fit. For the final model, we calculated the reduction in $R^2$ from removing each PROM from the model as a measure of the contribution of that variable to explaining variance in the WSAS outcome. Standardised effect estimates were produced to facilitate comparisons between the effect sizes of the PROMs, as they were each measured on different scales.

The analysis was conducted using a complete case approach, assuming data were missing at random (MAR) conditional on the variables included in the regression models. Comparisons were made between the demographic characteristics of the full sample of treatment-seeking patients and those providing a baseline WSAS measure to assess the potential for selection bias due to the exclusion of patients with missing WSAS scores.

### Secondary outcomes
#### WSAS domain score analysis
Secondary analysis was conducted to assess the extent to which the PROMs identified in the main analysis were associated with the individual domain scores of each of the five WSAS domains. The PROMs used in the multivariable logistic model were tested as explanatory variables in linear regression models for each of the five domains of ability to work, home management, social leisure activities, private leisure activities and close relationships. Models were adjusted for age and gender as in the primary analysis. Standardised estimates of effect size and change in adjusted $R^2$ values were calculated for each PROM in the multivariable model.

#### EQ-5D-5L analysis
Frequencies and proportions of patients reporting each dimension and level of EQ-5D-5L were calculated. Linear regression analysis of the EQ-5D index score was carried out to quantify the effect of patient demographics and PROMs on HRQoL. Multivariable linear regression models for the EQ-5D-5L analysis were developed, adopting the same model selection strategy used in the primary analysis.

**Table 2** Summary of PROMs and scores for users of the Living With Covid Recovery DHI

| PROM | Measures | Completed (n) | Mean (SD) | Threshold values (in each threshold category, n (%)) |
|---|---|---|---|---|
| WSAS Primary outcome | Functional limitations of the patient; higher scores indicate greater functional impairment. Range 0–40 | 2627 | 20.6 (9.9) | <10: subclinical (394 (15.0)) 10–19: significant (843 (32.1)) >20: moderately severe (1390 (52.9)) |
| Ability to work* | Functional limitations within domains Subscale range 0–8 0: not at all affected 8: very severely affected | 2621 | 4.6 (2.4) | |
| Home management | | 2627 | 4.2 (2.2) | |
| Social leisure activities | | 2627 | 4.0 (2.2) | |
| Private leisure activities | | 2627 | 4.7 (2.3) | |
| Close relationships | | 2627 | 3.0 (2.4) | |
| EQ-5D (EQ-5D-5L) Secondary outcome | A standardised measure of health status; index scores range from 0 (equivalent to dead) to 1 (full health); negative values are possible. | 2633 | 0.54 (0.27) Median: 0.60 (IQR 0.41–0.71) | |
| Explanatory variables | | | | |
| FACIT-F | Self-reported fatigue and its impact on daily activities and function. Higher scores indicate less fatigue. Range: 0–52 | 2890 | 19.6 (10.1) | <30: impairment (2418 (83.7)) ≥30: no impairment (472 (16.3)) |
| FACIT-F (Reversed Scale) Scale reversed in results to aid interpretation | Higher scores indicate greater fatigue. Range 0–52 | 2890 | 32.4 (10.1) | ≤22: no impairment (472 (16.3)) >22: impairment (2418 (83.7)) |
| GAD-7 | Screening tool and severity measure for anxiety Range 0–21 | 2774 | 9.0 (5.9) | <4: no anxiety (715 (25.8)) 5–9: mild anxiety (870 (31.4)) 10–14: moderate anxiety (591 (21.3)) ≥15: severe anxiety (598 (21.6)) |
| PHQ-8 | A valid diagnostic and severity measure for current depressive disorders Range 0–24 | 2661 | 11.8 (6.0) | <10: no depression (1034 (38.9)) ≥10: clinical depression (1627 (61.1)) |
| Dyspnoea-12 | Overall score of breathlessness impact, with higher scores corresponding to greater severity Range 0–36 | 2656 | 12.0 (9.3) | No threshold values |
| MRC Dyspnoea Scale, median (IQR) | Degree of breathlessness related to activity Range 1–5 | 2607 | 2 (2,3) | 1: mild (262 (10.1)) 2–3: moderate (1800 (69.0)) 4–5: severe (545 (20.9)) |
| PDQ-5 | Degree to which individuals perceive themselves as experiencing cognitive difficulties Range 0–20 | 2783 | 12.3 (4.3) | ≤8: minimal (519 (18.7)) 9–14: moderate (1346 (48.4)) ≥15: severe (918 (33.0)) |

Overall mean (SD) and number (%) within each threshold category are reported.
*Reduced number of completed answers as patients who had retired or chose not to work did not need to answer this question.
FACIT-F, Functional Assessment of Chronic Illness Therapy–Fatigue; GAD-7, Generalised Anxiety Disorder Scale, Seven-Item; MRC, Medical Research Council; PDQ-5, Perceived Deficits Questionnaire, Five-Item Version; PHQ-8, Patient Health Questionnaire–Eight Item Depression Scale; PROM, patient-reported outcome measure; WSAS, Work and Social Adjustment Scale.

*Working days lost due to PCS*
Additionally, LWCR users were asked to complete a study-specific questionnaire to capture data on the number of working days lost in the 28 days prior to questionnaire completion. Users were asked "In the last 4 weeks how many days off work (sick leave) have you taken due to COVID-19 and/or rehabilitation." The correlation between the number of working days lost and the WSAS 'work' domain was estimated.

All analyses were carried out in Stata V.17.0.

**Patient and public involvement**
This study had substantial PPI involvement with co-investigator (JB), steering group (JB, KB), individual work package management groups and an overall PPI Advisory Group. The feedback from PPI at an early stage was essential in determining the PROMs chosen in the study and the primary outcome measure of the WSAS.[20]

**Table 3** WSAS multivariable model for different patient characteristics and PROM scores (N=2556)

| Patient characteristics | | OR (95% CI) | P value | Reduction in $R^2$ (full model $R^2$=0.529) | Standardised effect size |
|---|---|---|---|---|---|
| Age | 18–29 | Reference | | | |
| | 30–39 | 1.18 (0.78 to 1.77) | 0.441 | | |
| | 40–49 | 0.90 (0.62 to 1.32) | 0.603 | | |
| | 50–59 | 0.62 (0.42 to 0.90) | 0.011 | | |
| | 60–69 | 0.55 (0.35 to 0.85) | 0.008 | | |
| | 70 and over | 0.26 (0.12 to 0.59) | 0.001 | | |
| Gender | Male | Reference | | | |
| | Female | 0.83 (0.66 to 1.05) | 0.115 | | |
| | Non-binary | 0.25 (0.05 to 1.17) | 0.078 | | |
| PROMs | FACIT-F (Reversed Scale) High values indicate greater fatigue. | 1.16 (1.14 to 1.18) | <0.001 | 0.179 | 4.47 |
| | PHQ-8 High values indicate more severe depression. | 1.05 (1.03 to 1.08) | <0.001 | 0.009 | 1.37 |
| | PDQ-5 High values indicate more perceived deficits. | 1.06 (1.03 to 1.09) | <0.001 | 0.009 | 1.29 |

FACIT-F, Functional Assessment of Chronic Illness Therapy–Fatigue; PDQ-5, Perceived Deficits Questionnaire, Five-Item Version; PHQ-8, Patient Health Questionnaire–Eight Item Depression Scale; PROM, patient-reported outcome measure; WSAS, Work and Social Adjustment Scale.

## RESULTS

### Patient demographics

The study included 3754 treatment-seeking patients with PCS with a mean age of 47.7 (SD 12.3) years, and 3541 (94.4%) being of working age (18–65) from across 31 clinics in the UK. The population were 71% (n=2675) female and 87% (n=2414) of White ethnicity (table 1) and skewed towards affluence, with 11% (n=289) from the most deprived quintile and 24% (n=642) from the least deprived. Just over a half (n=1466, 53%) were educated to degree level or higher. Similar patient characteristics were seen in those who completed the WSAS and EQ-5D PROMs compared with the overall sample of patients using the app (table 1).

### Functional impairment and quality of life of the treatment-seeking PCS population

#### Functional impairment

Characteristics of patients who completed the WSAS PROM were similar to those of all users of the LWCR DHI (table 1). The population reported a very high degree of functional impairment (mean WSAS score of 20.6, n=2627), with over half the patients (53%) scoring above 20 in the moderately severe category (online supplemental appendix figure 1). Functional impairment was seen across all five of the WSAS domains; with the highest rates of functional impairment seen in the Social Leisure Activities and Ability to Work categories; mean scores 4.7 and 4.6, respectively. The least affected domain in patients with PCS was close relationships with a mean score of 3.0 (online supplemental appendix 1). Ethnicity was not a contributing factor to the WSAS score; ethnicity was not significant in the univariable analysis and was therefore dropped from subsequent models. In increasing order, the mean WSAS score across the ethnic groups was: Black African Caribbean or black British 17.3; White 20.6; Asian or Asian British: 21.2; Mixed or multiple ethnicity 23.1; and 17.5 in those who preferred not to provide their ethnicity.

### Health-related quality of life

EQ-5D data was completed by 2643 LWCR DHI users. Patients reported a large impact on HRQoL, with an average (median) EQ-5D index score of 0.60 (IQR 0.41 to 0.71) (online supplemental appendix figure 2).

Appendix 2 shows the number of respondents reporting a problem in each domain. The two domains of the EQ-5D most affected by PCS were pain/discomfort reported by 2542 (96.2%) and anxiety/depression reported by 2509 (95%). The least affected EQ-5D domain was usual activities, with 36% reporting no problems.

### Working days lost due to PCS

Half (n=1321/2600, 50.8%) of patients who completed the study-specific questionnaire reported losing one or more days from work in the previous month, with a fifth (20.3%) reporting between 20 and 28 working days lost, as shown in Online supplemental appendix 3. The correlation between the baseline WSAS work domain (score 0–8) and the number of working days lost was 0.52, showing moderate correlation.

### Severity of patient-reported symptoms

The LWCR DHI users were extremely fatigued, reporting a mean FACIT-F score of 19.6, well below the threshold value of 30 used in this study (FACIT-F (Reversed Scale)

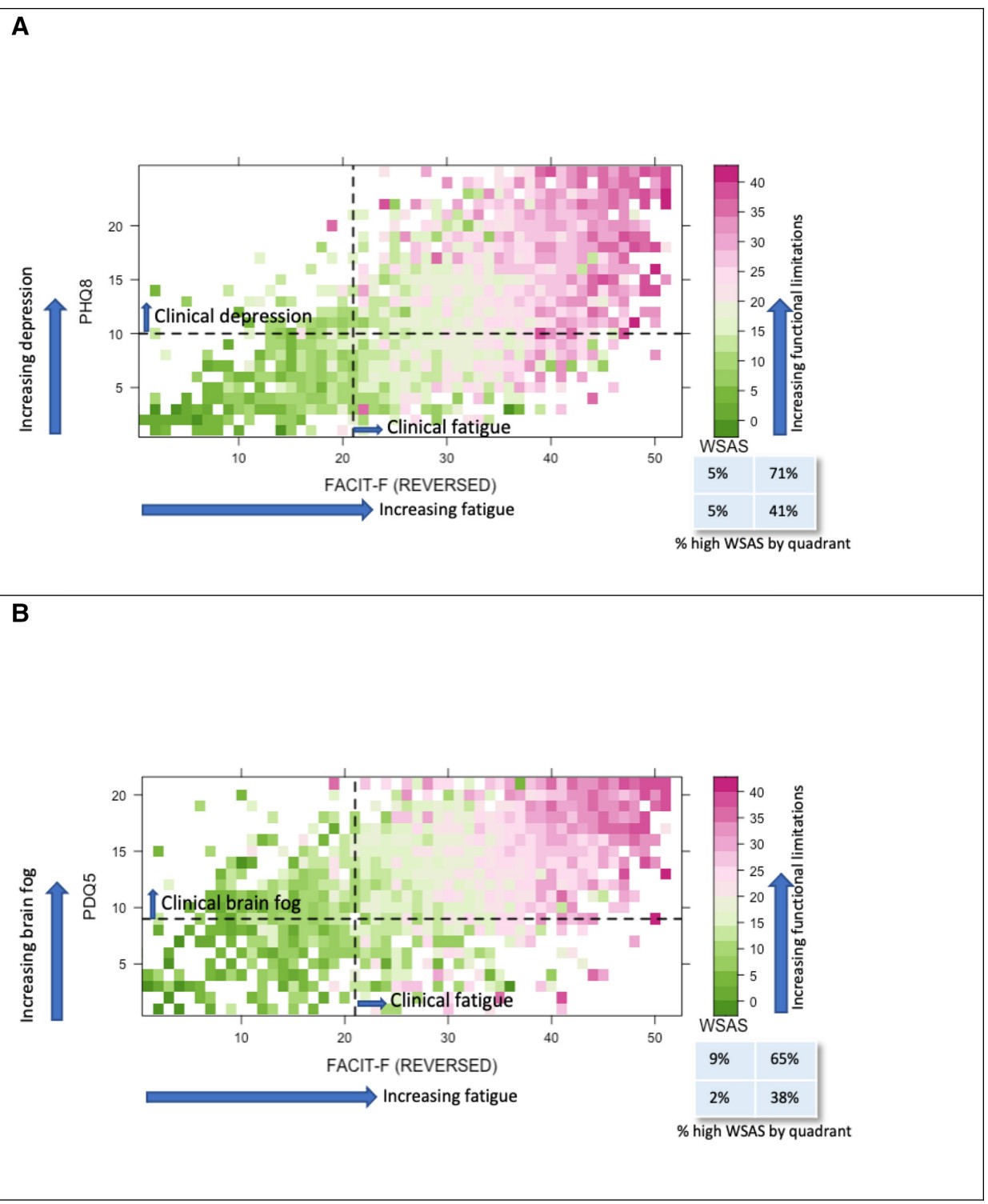

**Figure 1** (A) Heat map showing the distribution of each patient's (n=2502) WSAS scores (higher score representing an increase in functional limitations) compared with their corresponding fatigue levels FACIT-F (Reversed Scale) and depression (PHQ-8) levels. The dashed line represents the threshold values for significant fatigue on the x-axis and clinical depression on the y-axis.(B) Heat map showing the distribution of each patient's (n=2520) WSAS scores (higher score representing an increase in functional limitations) compared with their corresponding fatigue levels (FACIT-F (Reversed scale)) and brain fog (PDQ5) levels. The dashed line represents the threshold value for significant fatigue on the x-axis and moderate brain fog on the y-axis. FACIT-F, Functional Assessment of Chronic Illness Therapy–Fatigue; PHQ-8, Patient Health Questionnaire–Eight Item Depression Scale; WSAS, Work and Social Adjustment Scale.
Converted

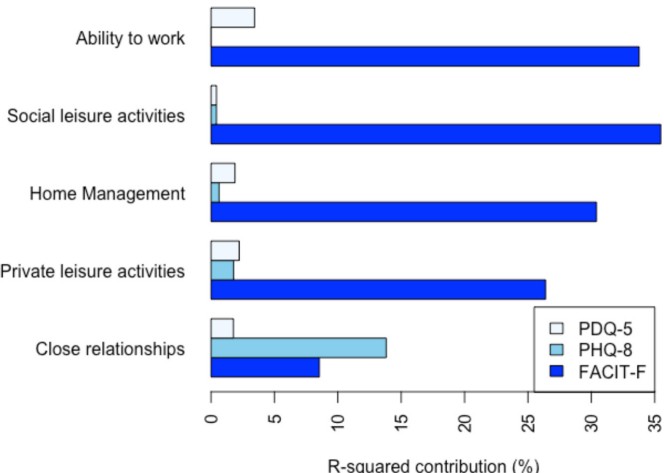

**Figure 2** Change in proportion of variation in WSAS explained (R-squared) when PROMs were removed from the linear regression models for each WSAS domain. FACIT-F, Functional Assessment of Chronic Illness Therapy–Fatigue; PDQ-5, Perceived Deficits Questionnaire, Five-Item Version; PHQ-8, Patient Health Questionnaire–Eight Item Depression Scale.

mean 32.4, threshold value of 22). Mental health was affected, with a mean GAD-7 score of 9 (corresponding to mild anxiety) and a mean PHQ-8 score of 11.8, meeting the clinical threshold for depression. Additionally, breathlessness was evident, with a mean Dyspnoea-12 score of 12 and median MRC Dyspnoea Scale score of 2 (IQR 2–3). The PCS population also reported moderate cognitive difficulties (brain fog) with a mean PDQ-5 score of 12 (table 2).

### Contribution of fatigue to functional impairment and HRQoL
#### Functional impairment
Fatigue, depression and cognitive impairment were significant predictors of a high WSAS (functional impairment) score. Fatigue was the strongest predictor of high WSAS, with a one-point increase in the reversed FACIT-F associated with an increase of 16% in the odds of a patient having a high WSAS score. When sequentially removing each PROM from the final multivariable model, the greatest contribution to reduction in $R^2$ (measure of goodness of fit of the statistical model) was attained by the removal of FACIT-F (33.8%), compared with a 1.7% reduction in $R^2$ for both PHQ-8 and PDQ-5) (table 3).

Figure 1 shows the heat map distribution of WSAS scores with almost all the high scores (denoted by pink squares) above the FACIT-F threshold for impairment. In contrast, the high WSAS scores are spread more evenly across both sides of the cognition and depression thresholds of 8 and 10 for PDQ-5 and PHQ-8, respectively (Figure 1A,B). FACIT-F also contributed strongly to the scores for each of the five WSAS domains, with PHQ-8 only making a substantive contribution, outperforming that of FACIT-F, in the 'close relationships' domain. The contribution of PDQ-5 was small compared with FACIT-F, with ability to work most associated with cognition (figure 2).

There was no significant difference in the functional impairment between genders, but a higher rate of functional impairment was seen in the younger age groups. The highest rate was seen in the 30–39 age group, compared with the reference age category of age 18 to 29 (OR 1.18, 95% CI 0.78 to 1.77; table 3).

### Health-related quality of life
Fatigue also contributed to the HRQoL of patients with PCS with the FACIT-F (Reversed Scale) being a significant predictor of the EQ-5D index score. FACIT-F (Reversed Scale) made the largest contribution to explaining variation in quality of life (change in $R^2$ of 8.4% compared with 5.6% for MRC Dyspnoea Scale, 3.1% for GAD-7, 1.7% for PHQ-8 and 0.5% for Dyspnoea-12. (online supplemental appendix 1).

### DISCUSSION
#### Principal findings
Treatment-seeking patients with post-COVID-19 consisting of mainly female, white, working age and well-educated people are experiencing striking levels of functional impairment and low HRQoL. This impairment is mainly driven by their fatigue level, causing significant impact on their ability to work and care for others.

The patients report levels of functional impairment worse than in several other known clinical cohorts, such as patients referred to Improving Access to Psychological Therapies (IAPT) services in the South West of the UK (mean score 18.8 at referral).[38] Functional impairment was worse than in patients who had a stroke (mean WSAS score of 16) and comparable to patients with Parkinson's disease (the mean WSAS scores ranged from 22.9 to 24.8), both debilitating neurological conditions.[39] Similarly, these patients report low HRQoL, with a mean EQ-5D score of 0.54 (SD 0.26), which compares poorly with patients with advanced/metastatic cancers.[40 41] For example, mean EQ-5D for stage IV lung cancer was between 0.66 and 0.84.[41] The results of the multivariable analysis show that fatigue is the strongest predictor of functional impairment (table 3) and HRQoL (online supplemental appendix 4). Our population of patients reported worse fatigue (mean score of FACIT-F 19.6) than patients with stroke (mean score 38), inflammatory bowel disease (mean score 38.9), end stage renal disease (mean score 39) and even anaemic cancer patients (mean score 24)[30 42–45] As well as patients reporting severe fatigue, they also report breathlessness, anxiety, depression and cognitive dysfunction.

This study is, to the best of our knowledge, the first reporting on functional limitations and HRQoL in PCS from a national population of patients referred for specialist rehabilitation. As such, they differ from other cohort studies which have followed up patients initially identified as hospitalised patients with acute COVID-19 (mean FACIT-F score 16.8) or through positive COVID-19 testing in the general public.[46] One study has recently

reported on a single-centre post-COVID-19 assessment clinic showing similar levels of fatigue but using a different measure (mean Fatigue Assessment Scale score 29) and inability of patients to work across hospitalised and non-hospitalised patients.[47] None of the other studies have reported on functional impairment using the WSAS, which measures the impact PCS is having on patients' normal daily activities.

This study enforces the recommendation for the use of a consistent set of outcome measures in studies in COVID-19. One such list of recommended variables is the International Consortium for Health Outcomes Measurement (ICHOM) set of patient-centred outcome measures for COVID-19, which recommends that research assesses functional status, quality of life and social functioning in addition to the typically reported measures of clinical outcomes, mental functioning and symptom reporting.[48 49] Additionally, consideration should be given to the interpretation of fatigue in patients with PCS, as advised by Sandler et al.[10] Patients may report fatigue when experiencing weakness, dyspnoea, cognitive dysfunction, somnolence or low mood.

### Strengths and limitations of this study

All the data collected in this study were recorded in real time by patients and were used by clinicians in their assessment and treatment. All PROMs used in the LWCR study were validated measures selected to provide the most reliable clinical information for patient benefit. Using these outcome measures allowed patient scores to be compared across disease types and with scores from other COVID-19 studies. This necessity for clinically led data collection led to substantial missing data, partly due to the DHI evolving to include new features over the reported period; patients who used the DHI later in its development were able to complete more PROMs. The primary reason for app usage and associated data collection was not for research; as a result, data on the severity of the initial disease or COVID-19 vaccination status were not collected within the app. Other studies have reported on the inconsistent relationship between severity of initial disease and severity of PCS[46 50]; therefore, we did not seek to capture further patient data from other sources.

Our chosen approach to the regression analysis was to use the observed data (a complete case approach), but we acknowledge that exclusion of the missing data may have introduced bias. An alternative approach to analysing data that are MAR would be to use multiple imputation, but it has been recommended that complete case analysis can be used as the primary analysis in situations where missing data are restricted to the dependent variable (we found very low levels of missing data in the explanatory variables when excluding patients with missing outcome data) and auxiliary variables have not been identified.[51]

Patients recruited to this study were sampled from the 31 specialist post-COVID-19 clinics that had chosen to use the LWCR DHI at the time of data extraction.

Our sample is representative of the patients who are seen in PCS clinics nationally. The data may not be representative of all patients with long COVID or PCS as many of these patients are not seen in a PCS clinic for a variety of reasons. This can be noted in the patient demographics, which shows that the majority of our patients are white, affluent and well-educated people. These patients are more likely to seek and obtain help than their counterparts.

This study has implications for the targeting of limited resources to effectively address functional limitations from PCS. Of particular concern is the large proportion of working age women in our study population, people who contribute substantially to the healthcare, social care and informal care sectors[52] at a time when these sectors are already under duress.[53] PCS is clearly a multifactorial disease affecting physical and mental well-being, but post-COVID-19 assessment services should consider focusing on assessing and treating fatigue to maximise the recovery and return to work in this large cohort of patients. Further work is needed to explore the recovery trajectories of this cohort over time and whether fatigue continues to predict functional impairment and low HRQoL over time.

### CONCLUSION

In this first UK national study reporting clinical symptoms from patients referred for assessment and treatment of PCS, we demonstrate high levels of functional impairment and low HRQoL. Fatigue appears to be the symptom most strongly associated with functional impairment. Currently, clinical services lack evidence-based approaches in treating patients experiencing fatigue related to PCS with no standard rehabilitation pathway.[11–14] This requires further targeted research. Our future work to explore the recovery trajectory of patients using the LWCR DHI may help to establish the extent to which WSAS and other PROMs are sensitive to changes in the health of a patient with PCS. This work can contribute to the identification of PROMs best suited for use in assessing, managing and treating patients with PCS, both digitally and in face-to-face appointments.

**Author affiliations**
[1]Department of Health and Community Sciences (Medical School), University of Exeter, Exeter, UK
[2]Department of Primary Care and Population Health, University College London, London, UK
[3]General Medicine, Whittington Health NHS Trust, London, UK
[4]Primary Care and Population Health, University College London, London, UK
[5]Research Department of Primary Care and Population Health, University College London, London, UK
[6]UCLIC, Department of Computer Science, University College London, London, UK
[7]Psychology, University of Southampton, Southampton, UK
[8]UCL Respiratory, University College London, London, UK
[9]Department of Respiratory Medicine, Barts Health NHS Trust, London, UK
[10]Respiratory Medicine, Barts Health NHS Trust, London, UK
[11]10 Queen Street Place, London, EC4R 1AG, Living With Ltd, London, UK

[12]Department of Applied Health Research, University College London, London, UK
[13]University College London, London, UK

**Collaborators** Living With Covid Recovery Collaboration: Julia Bindman, Ann Blandford, Katherine Bradbury, Tahreem Chaudhry, Encrico Costanza, Delmiro Fernandez-Reyes, Charlotte Foster, Henry Goodfellow, Manuel Gomes, Fiona Hamilton, Melissa Heightman, William Henley, John Hurst, Hannah Hylton, Stuart Linke, Elizabeth Murray, Rachel Okin, Paul Pfeffer, William Ricketts, Heidi Ridsdale, Chris Robson, Jane Simpson, Richa Singh, Fiona Stevenson, David Sunkersing, Sarah Walker, Jiunn Wang, Jonathan Waywell and Living With Covid Recovery Patient and Public Involvement Group.

**Contributors** EM and HG were responsible for the concept of the Living With Covid Recovery study. HG is the guarantor. SW was the first author of the manuscript and revised it after the review of the wider study team. SW, WH, HG and MG advised on the appropriate statistical design. SW and WH carried out the statistical analysis for the study. PP supported SW in preparing the paper for publication, including performing the literature search and drafting parts of the manuscript. SW, HG, PP, EM, JB, AB, KB, BC, FLH, JRH, HH, SL, PP, WR, CR, FAS, DS, JW, MG and WH contributed to the study design, reporting and review of the paper in steering comittee meetings and reviewed the paper prior to submission.

**Funding** This study is funded by the National Institute for Health and Care Research (NIHR) Cross-programme (HS and DR) COVID-19 (project reference NIHR132243). The views expressed are those of the authors and not necessarily those of the NIHR or the Department of Health and Social Care. This report is independent research supported by the NIHR ARC North Thames, NIHR ARC Wessex and NIHR ARC West. For the purpose of open access, the authors applied a creative commons attribution licence to any author accepted manuscript version arising.

**Competing interests** JB reports payments from University College London for working with the patient and public involvement group to prepare content for the digital health intervention since May 2020. KB's research portfolio is partly funded by National Institute for Health &and Care Research (NIHR) Applied Research Collaboration Wessex. HG reports working as a clinical safety officer for Living With Ltd. JRH reports receiving personal fees and fees to institution for honorariums and consultancy payments from AstraZeneca, Boehringer Ingelheim, Chiesi, GlaxoSmithKline and Takeda, and also sponsorship for attending meetings from AstraZeneca and GlaxoSmithKline. HH reports payment from the University of East London for providing a lecture on long COVID and COVID-19 recovery in February 2021. SL reports grants from NIHR in which the payment was made to Camden and Islington NHS Trust between the period of October–September 2022. PEP reports grants from the Medical Research Council and NIHR outside the submitted work. All other authors declare no competing interests.

**Patient and public involvement** Patients and/or the public were involved in the design, conduct, reporting or dissemination plans of this research. Refer to the Methods section for further details.

**Patient consent for publication** Not applicable.

**Ethics approval** This study involves human participants and was approved by the East Midlands–Derby Research Ethics Committee (reference 288199). Data were routinely collected in the Living With Digital Health Interface as part of clinical care. Patient consent was not required.

**Provenance and peer review** Not commissioned; externally peer reviewed.

**Data availability statement** Data are available upon reasonable request. To request access to the underlying research data, please contact Professor Fiona Stevenson: f.stevenson@ucl.ac.uk

**ORCID iDs**
Sarah Walker http://orcid.org/0000-0003-4201-1093
Henry Goodfellow http://orcid.org/0000-0002-0979-2839
Elizabeth Murray http://orcid.org/0000-0002-8932-3695
Ann Blandford http://orcid.org/0000-0002-3198-7122
John R Hurst http://orcid.org/0000-0002-7246-6040
William Ricketts http://orcid.org/0000-0002-0475-0744
William Henley http://orcid.org/0000-0001-6273-2619

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
