## [Reviewer comments · BMJ Open]

ARTICLE DETAILS

TITLE (PROVISIONAL)	The impact of fatigue as the primary determinant of functional limitations amongst patients with Post-COVID syndrome: a cross-sectional observational study
AUTHORS	Walker, Sarah; Goodfellow, Henry; Pookarnjanamorakot, Patra; Murray, Elizabeth; Bindman, Julia; Blandford, Ann; Bradbury, Katherine; Cooper, Belinda; Hamilton, Fiona; Hurst, John; Hylton, Hannah; Linke, Stuart; Pfeffer, Paul; Ricketts, William; Robson, Chris; Stevenson, Fiona; Sunkersing, David; Wang, Jiunn; Gomes, Manuel; Henley, William; Collaboration, LivingWith Covid Recovery

VERSION 1 – REVIEW

REVIEWER	Sahin, Mustafa Ondokuz Mayıs University
REVIEW RETURNED	12-Nov-2022

GENERAL COMMENTS	I have evaluated the manuscript titled "The impact of fatigue as the primary determinant of functional limitations amongst patients with Post-COVID syndrome: a cross-sectional observational study". The researchers have focused on an important issue. The manuscript fulfills all the scientific standards necessary to be published. The title explains the content of the manuscript well and the abstract includes necessary and sufficient data. Results and Discussion are sufficient and well-organized. Further statistical analyses are not required. In fact, the analyses made by the researchers are sufficient. The language of the manuscript is good enough to understand and there are not any spelling or punctuation mistakes. The manuscript has sufficient quality and originality to be published in BMJ Open. I do not think that the manuscript needs revisions. I think it can be considered for publication in the journal.
---

REVIEWER	Vyas, Jui Cardiff University, Centre for Medical Education
REVIEW RETURNED	23-Dec-2022

GENERAL COMMENTS	This is a very comprehensive and well written study which has studied the impact of POST-COVID syndrome in a substantial number of patients. The finding that fatigue appears to be the most important symptom associated functional impairment is important and early targeted intervention based on the findings of the study may have a positive impact on the NHS and directly be of benefit to economic recovery. I have a few minor changes to recommend:
--

	Methods: Intervention 1. Spell out WSAS in the Intervention section where it is mentioned first rather than in the outcome section. Outcomes: 2. Secondary outcome: EQ-5D-5L Perhaps insert a sentence on the coding for the 5 levels for each dimension as a reader not familiar with the questionnaire may not understand the 5 digit coding for the EQ-5D health state. This will also enable better understanding in the result section. Explanatory variables: 3. IMD has been spelled out twice. Only use the acronym after the first instance. The methods section does not make clear how 13 questionnaires have been used. Information has been provided on: 1WSAS 1EQ-5D 1Demographic questionnaire 6 PROMS(including 2 for dyspnoea). 4. What are the other 4 questionnaires, please elaborate. 5. Where applicable, for the questionnaires especially the PROMS, please provide the recall period (eg Today for EQ-5D, two weeks for GAD 7 etc) 6. Are there any references for the FACIT-F (reversed scale)? 7. Results: As you have stratified the ethnicities in the results (Table 1), it would be interesting to know if there were any differences in the outcomes between the different ethnicities and whether there is any explanation for differences (if any)
--	--

VERSION 1 – AUTHOR RESPONSE

Reviewer 1	No comments	No action required
Reviewer 2	Methods: Intervention 1. Spell out WSAS in the Intervention section where it is mentioned first rather than in the outcome section.	Thank you for pointing out this oversight. Acronym has now been defined in Intervention section on page 6, (WSAS) removed from Primary Outcome section on page 7. We have left the full name in the Primary Outcome section for ease of reading but continue to use WSAS throughout the remainder of the document.
Reviewer 2	Outcomes: 2. Secondary outcome: EQ-5D-5L Perhaps insert a sentence on the coding for the 5 levels for each dimension as a reader not familiar with the questionnaire may not understand the 5 digit coding for	Thank you for highlighting this. We have made some edits to give clarity on this. The previous text read: For each dimension, there are 5 possible responses (no problems, slight problems, moderate problems,

	the EQ-5D health state. This will also enable better understanding in the result section.	severe problems, unable to/extreme problems). We have now added additional text to include the scoring within this section. It now reads as: For each dimension, there are 5 possible responses (level 1: no problems, level 2: slight problems, level 3: moderate problems, level 4: severe problems, level 5: unable to/extreme problems). Additionally, these levels have been added to Table 2. In order to clarify that the EQ-5D analysis relates to the index score (rather than the VAS score), the final sentence in the Secondary Outcomes section has been edited. Tracked changes has been used so this can be seen by the Editorial team.
Reviewer 2	Explanatory variables: 3. IMD has been spelled out twice. Only use the acronym after the first instance.	Thank you. This has been edited.
Reviewer 2	The methods section does not make clear how 13 questionnaires have been used. Information has been provided on: 1 WSAS 1 EQ-5D 1 Demographic questionnaire 6 PROMS (including 2 for dyspnoea). 4. What are the other 4 questionnaires, please elaborate.	We have revised this section as there was sufficient clarity between the questionnaires (in which the PROMs were collected) in the DHI and those reported in this paper. There were 4 questionnaires which aren't related to PROMs and capture patient demographics, health service use or are 2 additional measures not used in this study (neither of these latter two are validated measures). The text has been amended from: It contains 13 (11 validated) patient-reported outcome measures (PROMs) in the form of validated questionnaires completed by patients as part of their clinical care. Seven related to symptoms and one related to each of patient demographics, functional ability, quality of life and health service use. To now read as: The DHI contains 12 (8 validated) patient-reported outcome measures (PROMs) in the form of validated questionnaires completed by patients as part of their clinical care. In this study, we use 10 of these (8

		validated). Six are related to symptoms and one related to each of patient demographics (unvalidated), functional ability, quality of life and health service use (unvalidated).
Reviewer 2	5.Where applicable, for the questionnaires especially the PROMS, please provide the recall period (eg Today for EQ-5D, two weeks for GAD 7 etc)	This is a useful addition. When the outcome measures and PROMs are defined, the recall period has been added.
Reviewer 2	6.Are there any references for the FACIT-F (reversed scale)?	We are not aware of a reference specifically related to this. Reversing the scale used for the score is a method often used in order to aid interpretation of results where differing scales are used whereby some scales have high scores to show positive results and vice-versa.
Reviewer 2	7. Results: As you have stratified the ethnicities in the results (Table 1), it would be interesting to know if there were any differences in the outcomes between the different ethnicities and whether there is any explanation for differences (if any)	In the Functional impairment section in the Results (page 14), we have added the following text to explain why ethnicity is not in the multivariable model for WSAS. Data for the effect of ethnicity on EQ-5D is available in Appendix 4 but not commented on: Ethnicity was not a contributing factor to the WSAS score; ethnicity was not significant in the univariable analysis and was therefore dropped from subsequent models. In increasing order, the mean WSAS score across the ethnic groups was: Mixed or multiple ethnic groups: 9.7; White: 9.8; Asian or Asian British: 10.4; Other ethnic group: 10.4; Black, Black British, Caribbean or African 10.7 and 12.8 in those who preferred not to provide their ethnicity.

VERSION 2 – REVIEW

REVIEWER	Vyas, Jui Cardiff University, Centre for Medical Education
REVIEW RETURNED	28-Feb-2023
GENERAL COMMENTS	Thank you for clarifying the queries, no further revisions are necessary.